# Dirty Jobs: Macrophages at the Heart of Cardiovascular Disease

**DOI:** 10.3390/biomedicines10071579

**Published:** 2022-07-02

**Authors:** Travis W. Stevens, Fatimah K. Khalaf, Sophia Soehnlen, Prajwal Hegde, Kyle Storm, Chandramohan Meenakshisundaram, Lance D. Dworkin, Deepak Malhotra, Steven T. Haller, David J. Kennedy, Prabhatchandra Dube

**Affiliations:** 1Department of Medicine, University of Toledo College of Medicine and Life Sciences, Toledo, OH 43606, USA; tsteven16@rockets.utoledo.edu (T.W.S.); kareem.khalaf@utoledo.edu (F.K.K.); sophia.soehnlen@rockets.utoledo.edu (S.S.); prajwal.hegde@rockets.utoledo.edu (P.H.); kyle.storm@rockets.utoledo.edu (K.S.); chandramohan.meenakshisundaram@utoledo.edu (C.M.); lance.dworkin@utoledo.edu (L.D.D.); deepak.malhotra@utoledo.edu (D.M.); steven.haller@utoledo.edu (S.T.H.); 2Department of Clinical Pharmacy, University of Alkafeel, Najaf 54001, Iraq

**Keywords:** macrophages, cardiovascular disease, CVD, atherosclerosis, heart failure

## Abstract

Cardiovascular disease (CVD) is one of the greatest public health concerns and is the leading cause of morbidity and mortality in the United States and worldwide. CVD is a broad yet complex term referring to numerous heart and vascular conditions, all with varying pathologies. Macrophages are one of the key factors in the development of these conditions. Macrophages play diverse roles in the maintenance of cardiovascular homeostasis, and an imbalance of these mechanisms contributes to the development of CVD. In the current review, we provide an in-depth analysis of the diversity of macrophages, their roles in maintaining tissue homeostasis within the heart and vasculature, and the mechanisms through which imbalances in homeostasis may lead to CVD. Through this review, we aim to highlight the potential importance of macrophages in the identification of preventative, diagnostic, and therapeutic strategies for patients with CVD.

## 1. Introduction

Cardiovascular disease (CVD) encompasses a constellation of pathologies and diseases, including atherosclerosis, heart failure, cardiomyopathy, valvular heart disease, and arrhythmia. Despite vast research efforts and public health campaigns, CVD remains the world’s largest cause of mortality, accounting for nearly one-third of health-related deaths in the United States and other major countries [1,2,3]. Lifestyle, diet, and metabolic dysfunctions are well known to be the primary risk factors of CVD, and have been found to increase the prevalence of comorbidities such as obesity, hypertension, and hyperglycemia, as well as dyslipidemia [4,5]. Other biological factors have also been linked to an increased risk of CVD. Vascular calcification, which is often associated with chronic kidney disease (CKD), has been shown to worsen atherosclerosis and increase the risk of cardiac events, including heart attack and stroke, which has also made it a valuable predictor of CVD [6,7,8]. The unifying element among all of these risk factors is the increased presence of arterial and tissue inflammation, while the increased presence of pro-inflammatory markers in circulation is often associated with CVD and decreased tissue regeneration [9,10]. It is important to understand the body’s natural regulation of tissue inflammatory responses to gain better insight into the possible effective treatments of CVD conditions.

The inflammatory responses within tissues are now better understood to be a regulated and balanced process between the activity of macrophages in both the tissue itself as well as the blood circulation [11,12]. Macrophages are immune system cells that are known for their defense against infectious pathogenic agents, as well as their maintenance of homeostasis in tissue that has been damaged during disease [12]. Traditionally, macrophages were defined based on their inflammatory activity, which categorized them as either M1 pro-inflammatory macrophages or M2 anti-inflammatory macrophages (Figure 1). M1 macrophages can induce inflammatory responses in tissue through the secretion of pro-inflammatory cytokines and the recruitment of other inflammatory stimulating cells. In contrast, the M2 macrophages relieve inflammation through the signaling apoptosis of affected cells and by activating anti-inflammatory processes [12,13]. Although this simplified classification remains true, it does not consider the varying plasticity of macrophages, which have been shown to possess a variety of phenotypic characteristics depending on the resident tissue and the disease that is present. Both past and recent studies have also relied on the usage of electron microscopy to visualize macrophages and identify specific cell surface receptors that provide insight into their function [14,15]. It is now better understood that macrophages display specialized functions that work to maintain homeostasis in their resident tissues and to offset the effects of tissue-damaging pathologies.

In this review, we discuss the role of macrophages in maintaining cardiac tissue homeostasis and the major cellular mechanisms of macrophages in the presence of various CVD-related conditions, including heart failure, atherosclerosis, and cardiomyopathy.

## 2. Cardiac Macrophage Plasticity

Macrophages reside in all tissues and are important in development, immunity, tissue repair, and homeostasis [14]. Traditionally, macrophages have been classified based on the presence of cell surface markers and their role in inflammatory responses. As mentioned, this classification does not accurately reflect the plasticity and varying functionality of macrophages in different tissues. Macrophages are a diverse set of immune cells that possess a varying range of phenotypes and functions that are often determined by the specific tissue where they reside [14,16,17]. A more recent method of classifying macrophages relies on their developmental origins to indicate their location and function [12]. Studies in the last eight years have shown that many tissue-specific macrophages originate during the embryonic stage of development rather than from circulating monocytes, which originate from the bone marrow [15,18]. In both human and mouse hearts, macrophages are classified using both cell surface markers and their lineage.

Cardiac-specific macrophages are defined by their expression of C-C motif chemokine receptor 2 (CCR2), and are often referred to as CCR2^−^ and CCR2^+^ resident macrophages [14,15,19]. These resident macrophages are incorporated into the heart at various stages of development and possess their own specialized functions to maintain cardiac homeostasis and respond to injury. CCR2^−^ residents originate from yolk-sac and fetal liver progenitors during the embryonic stage of heart development and exhibit low expression levels of CCR2 [16,18,20]. CCR2^−^ macrophages are the primary macrophages associated with cardiomyocytes and make up the majority of macrophages in the heart. The maintenance of CCR2^−^ macrophages has been shown to operate independently of the input by circulating blood monocytes from the bone marrow, but instead CCR2^−^ populations are maintained via local self-proliferation [20,21]. The origin of CCR2^+^ residents differs greatly from that of CCR2^−^ residents in that they originate from bone-marrow-derived monocytes and retain their expression of CCR2 [15,18]. CCR2^+^ residents are first incorporated into the heart as a subset with low MHC II expression during the embryonic stage of development. However, during the neonatal stage, further subsets are incorporated with varying MHC II expression levels. In contrast to the maintenance of CCR2^−^ macrophages, CCR2^+^ macrophages require the input of blood monocytes to differentiate into CCR2^+^ monocytes [21]. In addition to the two classes of resident macrophages, both adult and neonatal hearts utilize the recruited macrophages to respond to tissue injury. The recruited macrophages are bone-marrow-derived monocytes that are often recruited by CCR2^+^ residents following cardiac injury [22].

## 3. Macrophage Function in Heart Homeostasis

The functions of the cardiac residents and recruited macrophages typically operate in opposition to maintain cardiac homeostasis. The CCR2^+^ macrophages seem to be primarily pro-inflammatory and seem to function through the delivery of pro-inflammatory interleukins via the NPLR3 pathway following tissue injury [15,23,24]. In response to cardiac injury, such as in atherosclerosis, the CCR2^+^ residents stimulate the recruitment of blood monocytes and neutrophils to enhance the inflammatory response. In addition to this inflammatory action, these resident macrophages also possess the ability to act as antigen presenters to initiate T-lymphocyte responses [21,25]. During the steady state, the specific functions of CCR2^+^ macrophages are still not entirely known; however, it appears that the activity of these residents may be controlled by the presence of mitochondrial DNA, which would be absent in a normal healthy cardiomyocyte [24,26]. Blood monocytes that are recruited by CCR2^+^ macrophages act to further stimulate pro-inflammatory responses. These recruited macrophages further the actions of CCR2^+^ macrophages through leukocyte recruitment, the promotion of oxidative damage, and the phagocytosis of damaged cardiac cells [15,19,27,28]. In contrast to the actions of CCR2^+^, which can be tissue-damaging, the recruited macrophages have also been shown to promote angiogenesis as a possible repair mechanism, although whether this is done in conjunction with CCR2^−^ macrophages or as a separate mechanism is not certain [28].

The CCR2^−^ macrophages typically function in contrast to the mechanisms of the CCR2^+^ and recruited macrophages, and play a major role in the development of the heart during embryonic development [18,20,29]. In the prenatal heart, CCR2^−^ macrophages exhibit typical anti-inflammatory M2 activity, as well as cardiac tissue generation and angiogenesis activities. In neonatal and adult hearts, CCR2^−^ macrophages continue to exhibit M2 activity, in contrast to the M1 activity of CCR2^+^ [30,31]. In adult human and mouse hearts, subsets of CCR2^−^ macrophages have shown continued meditation of angiogenesis and the expression of insulin-like growth factor (IGF-1) [18,23,32]. As a further contrast to CCR2^+^‘s activity, CCR2^−^ macrophages have shown the potential to inhibit the recruitment of blood monocytes by CCR2^+^ macrophages, as well as possibly to expel pro-inflammatory material from cardiomyocytes [23]. Interestingly, the subsets of CCR2^−^ macrophages also seem to express some ability to promote T-lymphocyte responses via antigen presentation during tissue stress, similar to CCR2^+^ macrophages [21,25]. During the steady state, CCR2^−^ residents have shown regenerative abilities following tissue damage or an inflammatory response, and have also been shown to possibly mediate the conduction of electrical synapses during homeostasis [33] (Figure 2).

## 4. Macrophages and Heart Failure

Heart failure is another common pathological consequence of CVD and has been a major concern in public health, as it affects roughly five million people each year in the United States [1,34,35,36]. Heart failure can result from a variety of causes, including diabetes, valvular heart disease, high blood pressure, and atherosclerosis [1,36]. Irrespective of the cause, heart failure commonly results in fatigue, difficulty breathing, and often death. As the prevalence of heart failure rises worldwide, new treatments and therapeutic methods are being developed to counter the cellular mechanisms involved in the development of this disease and the subsequent mechanisms that follow [37].

As discussed throughout this review, macrophages are responsible for maintaining tissue homeostasis in the heart. This role is carried out through inflammatory mediation by M1 and M2 macrophages, the recruitment of inflammatory cytokines and neutrophils, as well as the repair and remodeling of cardiac tissue [11,13]. The exact roles of macrophages during heart failure are not entirely known; however, different pathways have been observed based on the form of heart failure that is present [38,39]. Heart failure is typically classified into three forms: heart failure with a preserved ejection fraction (HFpEF), heart failure with a reduced ejection fraction (HFrEF), and heart failure with a mid-range ejection fraction (HFmrEF) [39,40]. HRpEF is a diastolic dysfunction of the left ventricle, where the chamber is unable to take in appropriate amounts of blood from the left atrium and is often associated with metabolic disorders such as obesity, diabetes, and hypertension [41,42]. HRrEF is a systolic dysfunction of the left ventricle, where the chamber is unable to pump the appropriate amount of blood out to tissues and is often associated with heart specific conditions such as myocardial infarction, cardiomyopathies, and valvular heart disease [41,42]. HRmrEF occurs when the systolic output of the left ventricle falls between the levels of HFrEF and HRpEF, and is often comparable to HfrEF [41,43,44]. These forms differ in their progression and risk factors, as well as their levels of inflammation and fibrosis [40]. Macrophages are known to play important roles in the development of heart failure through their involvement in hypertension, renal calcification, and atherosclerosis, which are all risk factors of heart failure. In HRrEF and HRpEF, inflammation caused by the increased levels of macrophage-recruited cytokines has been linked to more severe outcomes [41,42,43,45,46]. In acute HFrEF, such as following myocardial infarction, a significant loss of cardiomyocytes from tissue damage initiates the inflammatory response via the production of inflammatory cytokines. Over time, this inflammation results in a reduction in resident macrophages in the affected area, meaning the resident CCR2^+^ macrophages recruit more circulating M2 macrophages and fibroblasts to begin the reparative process [22,46,47]. At the onset of the reparative process, the macrophages release mediators known as specialized proresolving mediators (SPMs), which further stimulate anti-inflammatory M2 activity [48]. Through this process, the inflammation is typically resolved within one to two weeks following the initial injury and after the clearance of dead cardiomyocytes and cellular debris by the macrophages [49,50]. Over time, the excess repetition of this process can lead to chronic HRrEF due to overcompensatory mechanisms that thin the myocardial wall and chambers. In the progression of chronic HRrEF, the macrophages have been shown to continue their proliferation, with an elevated abundance of CCR2^+^ macrophages associated with deteriorating systolic dysfunction in the left ventricle, although whether this is a cause or consequence of HfrEF is unclear [16,51]. The increased levels of CCR2^+^ macrophages have shown significant reductions in the clearance of necrotic cardiomyocytes, which further stimulate inflammatory responses and induces tissue damage [21]. In past studies, HRrEF patients have shown increased levels of IL-1β and tumor necrosis factor (TNF-α), which are both pro-inflammatory cytokines and are correlated with worse clinical outcomes [52,53]. These finding suggest that macrophages may have a direct role in the onset and progression of HFrEF via the IL-1β pathway.

The role of macrophages in HFpEF is not as well understood compared to HFrEF. While inflammation and fibrosis are known to play direct roles in the progression of HFpEF, the prominence of cardiac macrophages in HFpEF’s progression is less understood due to it being more correlated with systemic metabolic comorbidities than other forms of HF. Fibrosis is a key characteristic found in heart failure, particularly HFpEF [46,54,55,56,57,58,59]. The exact method of fibrosis in the myocardium during heart failure is not known; however, it is possible that the mechanism is similar to that seen in vascular fibrosis. Vascular fibrosis is commonly seen in obesity and metabolic syndrome, which are both commonly correlated to heart failure. Following the apoptosis of adipocytes, M1 macrophages are recruited to stimulate inflammation through matrix metalloproteinase (MMPs) and transforming growth factor-B (TGF-B) activity, which leads to injury of the vascular wall [60]. The overexpression of MMPs has been linked to the excess deposition of collagen in the vascular wall, leading to hardening. Similar mechanisms have been seen in cardiac tissue through the macrophage secretion of interleukin-10 [55,60]. IL-10 is typically an anti-inflammatory molecule that is released by M2 macrophages to initiate tissue repair; however, excess IL-10 leads to the production of osteopontin, proteinases, and MMPs, which are linked to collagen deposition. This deposition of collagen results in stiffness of the myocardial wall and prevents the heart from properly pumping blood to tissues [61,62].

The tissue-repairing mechanisms of macrophages may provide valuable insights in the development of cell-based therapies for heart failure patients. In both ischemic and non-ischemic heart failure, resident M2 macrophages have exhibited repair mechanisms of the myocardial wall, particularly through myeloid–epithelial reproductive tyrosine kinase (MER-TK), which is a macrophage receptor that is believed to aid in the destruction of damaged cells [63,64,65]. Thus, efforts have been placed into finding therapies that can properly balance M1 and M2 populations in the heart [66,67]. As shown in other forms of CVD, the macrophages present a complex heterogeneity, even within the same populations, and can vary in their responses and mechanisms [68]. The diverse responses between subpopulations of these M1/M2 populations and between the resident and recruited cardiac macrophages increases the complexity of therapies that may target these cell types. Therefore, further research into the mechanisms of macrophage regulation during and following heart failure is needed.

## 5. Macrophages and Atherosclerosis

The role of macrophages in the development and progression of atherosclerosis has been well defined over years of research. In short, the arteries that experience disturbed laminar flow, high stress, and injury are predisposed to the development of atherosclerosis due to the accumulation of apolipoprotein B-containing lipoproteins (apoB-LPs) at high-risk sites, as well as the adherence of platelets, macrophage chemotaxis, foam cell formation, and smooth muscle alterations [69,70,71]. The accumulation causes an inflammatory response cascade through the binding of monocytes to the luminal endothelium, which is one of the first indicators of early atherosclerosis [72]. Monocytes migrate into the arterial intima where they further differentiate into macrophages [73]. This is characterized by an increase in the CD68 antigen [74]. In fact, the specific targeting of the expression of chemokine receptors and cell adhesion molecules on monocytes in their mechanism of macrophage differentiation is a rapidly developing area of research for the treatment of atherosclerosis [75,76,77,78]. After differentiation, macrophages metabolize a variety of subendothelial components, such as aggregated and fused lipoproteins, leading to the formation of foam cells. In the late stages of atherosclerosis, the foam cells aggregate together to form an atheromatous core, where death occurs through necrosis (Figure 3). As the disease progresses, more foam cells accumulate around the necrotic core and a fibrous cap forms around the lesion [72,79]. This has provided attractive options to mechanistically target macrophages as a therapeutic approach in slowing the progression of atherosclerosis [80,81,82,83].

Interest has been drawn to the interplay of long non-coding RNA (IncRNA) in macrophages and the development of atherosclerosis due to emerging research on IncRNAs that suggests they have a strong connection to the regulatory pathways of microRNAs [84,85]. While the roles of IncRNAs are still poorly understood in macrophages, the recent research has displayed promising results [86]. The macrophage-associated atherosclerosis IncRNA sequence (MAARS) was recently identified and characterized [87]. In atherosclerotic LDLR-/- mice, MAARS was discovered to be a regulator of macrophage apoptosis, efferocytosis, and plaque necrosis through a direct interaction with the RNA-binding protein HuR [87]. The expression of MAARS increases by 270-fold alongside the progression of atherosclerosis, as well as decreases by 60% with atherosclerosis regression [87]. The knockdown of MAARS resulted in a 52% decrease in lesion formation [87,88]. Furthermore, Yu et al. suggest that the lncRNA kcnq1ot1 plays a role in atherosclerosis and found that the overexpression of kcnq10t1 caused a decrease in the expression of ATP binding cassette transporter A1 (ABCA1), a transmembrane protein that mediates the efflux of cholesterol in macrophages and miR-452-3p [89]. When further examining microRNA pathways in atherosclerosis, studies have identified the lncRNA myocardial-infarction-associated transcript (MIAT) in macrophages as a key player [90,91,92,93,94]. MIAT expression is elevated in the macrophages found in the necrotic core of atherosclerotic mouse models, as well as in the serum from patients with advanced atherosclerosis [95]. When MIAT was knocked down in mouse models, the macrophages displayed increased apoptotic cell clearance, reduced necrotic core formation, and increased plaque stability [95]. Moreover, another study demonstrated that MIAT expression was significantly correlated with a variety of indicators of macrophage activity, inflammatory cytokines, and growth factors, as well as vascular smooth muscle cell marker genes, a hallmark of atherosclerosis. MIAT also participated in the phenotypical switch of SMC to inflammatory macrophage-like cells that contributes to the progression of atherosclerosis and vascular inflammation [96].

The macrophage inflammatory immune response is one of the contributing factors of atherosclerotic plaque formation. Reducing the inflammatory response of macrophages has been associated with antiatherogenic properties, as displayed in the following studies. Macrophages express glucocorticoid receptors that can lead to anti-inflammatory properties. In order to elicit the anti-inflammatory properties, the synthetic glucocorticoids prednisone and prednisolone were administered to mice. The glucocorticoid-treated mice showed reduced triglyceride and cholesterol accumulation [97]. This finding supports the protective role of glucocorticoids in the inflammatory process of foam cell formation, and ultimately in the development of atherosclerosis [97,98]. Another known regulator of inflammation is ApoA-I, which counteracts the inflammatory response through binding to ABCA1 to export cholesterol out of macrophages. ApoA-I binding protein (AIBP) was also found to promote apoA-I binding to further inhibit the inflammatory response in the development of atherosclerosis [99]. Additionally, in the late stages of atherosclerosis, it has been demonstrated that macrophages have increased expression of filamin A (FLNA), which is an actin binding protein linked to cell architecture and signaling pathways [100,101]. The macrophages displayed impaired migration, proliferation, lipid uptake, and secretion of interleukin 6, an inflammatory marker, through filamin-A-dependent processes [102,103]. Inactivating FLNA in mice models also led to impaired macrophage signaling and function, accompanied by reduced atherosclerosis [100]. Transient receptor potential canonical 3 (TRPC3) in macrophages has been linked with various mechanisms in atherosclerosis disease progression [104]. In novel studies, mice with a macrophage-specific deficiency of transient receptor potential canonical 3 (TRPC3) displayed reduced stress-induced apoptosis, necrosis, and calcification in advanced atherosclerotic plaque [105,106,107]. With the advancements in the understanding of the different roles macrophages play in the development of atherosclerosis, therapeutic advances have been translated from bench to bedside, with promising early results [108].

It is known that the ABCA1- and G1-mediated cholesterol efflux from macrophages leads to anti-atherosclerotic outcomes. After clinical treatment with low and high doses of rosuvastatin, patients had lower plaque ABCA1 mRNA levels (although the protein levels were increased in the patients given the high dose) [109]. The rosuvastatin treatment was also associated with lower levels of miR-33b-5p, which is a known microRNA regulator of the ABCA1 gene [109]. Additionally, the therapeutic treatment with 10 mg of rosuvastatin plus 1800 mg/day of eicosapentaenoic acid (EPA) led to decreased serum LDL compared to patients on 2.5 mg/day of rosuvastatin at 12 months from baseline [107]. Another study demonstrated that intensive treatment of 10 mg/day of rosuvastatin plus 1800 mg/day of EPA decreased the progression of atherosclerosis in patients with neoatherosclerosis (NA) compared to the treatment group receiving 2.5 mg/day of rosuvastatin [110]. The findings were correlated with significant decreases in lipid index and macrophage grade along with an increase in 18-HEPE, which has been shown to inhibit macrophage-mediated inflammation [111,112]. The receptor Erv/Chemr23 regulates the uptake of oxidized low-density lipoproteins and phagocytosis through an 18-HEPE-derived lipid mediator. In murine models, the targeted deletion of Erv/Chemr23 led to proatherogenic signaling in macrophages through increased oxLDL and reduced phagocytosis, leading to an increased plaque size and necrotic core formation. This may further explain the implications of increased 18-HEPE in the previous study [111,113]. Furthermore, studies have been exploring the use of sonodynamic therapy (SDT), which involves the use of ultrasound to induce cell death through activating sono-sensitizers found in cells, as a non-invasive procedure for atherosclerotic plaque reduction [114]. The results of SDT with atorvastatin showed greater plaque shrinkage and lumen enlargement compared to atorvastatin alone. This was due to the increased macrophage apoptosis and enhanced efferocytosis as a result of the SDT [115].

## 6. Macrophages and Cardiomyopathy

Cardiomyopathy describes a range of disease manifestations and processes with varying etiologies that result in dysfunction of the heart muscle and an inability to pump blood. These etiologies (idiopathic, viral, genetic, hypertensive) have been well-described [116]. Until recently, however, the full role of macrophages in cardiomyopathy was not well-understood. In response to myocardial injury, both resident and non-resident macrophages can have an array of inflammatory or anti-inflammatory effects, depending on the tissue origin, etiology of the heart damage, and surrounding micro-environments [63,117]. Previous studies have demonstrated the importance of resident macrophages in the homeostasis and resolution of inflammation due to the phagocytotic nature of resident macrophages in clearing damaged cardiomyocytes [21]. Additionally, the depletion of resident macrophages has been shown to slow the remodeling and recovery from cardiac injury [118]. Logically, the expansion of these resident macrophages has also been shown to attenuate instances of cardiomyopathy [119]. Non-resident macrophages, distinguished from resident macrophages by the presence of CCR2, have been associated with inflammatory phenotypes, and previous studies have demonstrated that blocking non-resident macrophage migration into the heart results in a protective phenotype [21]. It has been suggested that the infiltration of inflammatory macrophage populations is also associated with a decline in the resident macrophage population [29], linking the two populations in pathogenesis. Further supporting this link is evidence suggesting that shifting the polarity of the macrophage from the M1 state (pro-inflammatory, associated with inflammatory cytokines such as IL-1β and IL-6) to the M2 state (anti-inflammatory, associated with anti-inflammatory cytokines such as IL-10) can be cardioprotective and can reduce tissue injury [63,120,121]. M1 macrophages dominate the initial response to injury among infiltrating monocytes, with an eventual shift to M2 macrophages to initiate repair [66]. While this offers a potentially exciting set of therapeutic targets, many other macrophage states exist, which vary based on the location in the heart tissue, requiring further phenotypic studies and delineations [122].

The balance between macrophage states and their roles dictates the response to ischemia and highlights the mechanism for macrophage involvement in the development of ischemic cardiomyopathy. However, macrophages have also been implicated in the response to non-ischemic cardiomyopathy. Liao et al. determined that in pressure overload hypertrophy models, M1 macrophage infiltration occurs late in the response to pressure overload hypertrophy and is associated with ventricular dysfunction [63]. Blocking M1 macrophage migration via CCR2 knockout models demonstrated some alleviation of this phenotype [63]. Additionally, Liao et al. noted that knocking-out KLF4 (i.e., eliminating resident macrophage expansion) resulted in a phenotype of dilated cardiomyopathy [63]. Interestingly they noted that unlike in ischemic cardiomyopathy, altering the M1/M2 polarization did not significantly change the phenotype involved in pressure overload hypertrophy (POH). This differed from the findings by Ren et. al, as they deployed the anti-inflammatory drug dapagliflozin in an attempt to mitigate macrophage polarization towards an inflammatory state, and found that it reduces the hypertrophy of cardiac muscle in response to pressure overload [123]. The work by Zhang et al. seemed to corroborate this, as they suggested through a bioinformatic approach that disruption in the function of CD163+/LYVE-1+ resident tissue macrophages is one of the most critical factors in the development of hypertrophic cardiomyopathy [124]. The deletion of LYVE-1 in mice was associated with poor cardiac function and repair, while the presence of LYVE-1 was associated with proper tissue homeostasis. Hence, the principle of macrophage polarization and the ability of macrophages to migrate this is critical in managing the response to ischemia.

Viral myocarditis is a condition that can lead to the development of dilated cardiomyopathy (DCM), and potentially leads to a significant disease burden [125]. Macrophages are known to be involved with the development of dilated cardiomyopathy due to their role in the chronic impact of viral myocarditis. Recently, with the COVID-19 pandemic, Guzik et al. suggested the ability of the SARS-CoV-2 virus to infect macrophages and other immune cells through the ACE2 receptor [126]. Upon infiltration of the heart tissue (as described earlier), these macrophages are thought to cause damage to cardiac tissue and subsequent myocarditis [126]. Chronic sequelae could then lead to DCM. The recent work by Xue et al. revealed new information regarding the manipulation of the macrophage polarization towards an M2 phenotype through the silencing of lncRNA MEG3 in mouse models [127]. This is thought to be through the modulation of miR-223, an miRNA involved in inflammatory processes that has previously been shown to regulate macrophage polarization [127]. Xue’s group showed that this pathway is important in the protection from Coxsackie B virus-induced myocarditis. Wu et al. noted that in Coxsackievirus B3 mouse models, macrophages with the α1β1-integrin receptor on their cell surface are critical in the development of viral myocarditis by binding the molecule Semaphorin7A and promoting an inflammatory phenotype [128]. These findings suggest the possibility of developing therapeutics that can mitigate the damage caused by macrophages in viral myocarditis and can reduce the emergence of DCM.

The general role of macrophages in cardiac inflammation and cardiomyopathy is not limited to cases of acute cardiac injury. Indeed, evidence exists for a role of macrophages in the pathogenesis of cardiomyopathies triggered by a low-grade, chronic level of inflammation. One such disease, Takotsubo cardiomyopathy, is traditionally associated with acute onset [129]. Indeed, Scally et al. recently reported high levels of macrophages and macrophage-mediated inflammation in patients with acute Takotsubo cardiomyopathy, implying a role for them in disease progression, despite the uncertainty regarding the true pathogenesis [130]. More interestingly, however, they reported that this inflammation may persist for as long as 5 months after the initial inciting event, with many pro-inflammatory cytokine markers remaining elevated in follow-up relative to controls [130]. Nishida et al. reported that although the mechanism has not been fully elucidated, the inflammation mediated by macrophages may contribute to the formation of metabolic cardiomyopathy and eventual systolic–diastolic dysfunction over a prolonged period [131]. This idea is supported by the work by Liu et al., who found that the actions of the SIRT1/miR-471-3p pathway led to the polarization of macrophages towards an M1 phenotype and contributed to diabetic cardiomyopathy [132]. SIRT1 is a histone deacetylase that was previously found to partially ameliorate the endothelial damage induced by diabetes, and these anti-inflammatory roles extend to macrophage polarization [132]. However, this is regulated by miR-471-3p, so the inhibition of miR-471-3p leads to polarization away from an M1 phenotype by freeing SIRT1 [132]. Cirrhotic cardiomyopathy, a condition of conjoined heart and liver dysfunction, can also manifest as a chronic condition with a role involving macrophages. Wiese et al. reported that in patients with cirrhotic cardiomyopathy, the levels of myocardial ECV (extracellular volume) and liver ECV (a measure of fibrosis) were elevated compared to healthy controls [133]. Importantly, they noted that macrophage activation markers such as sMR were increased in patients with higher degrees of cardiac fibrosis and liver fibrosis, suggesting a correlation between the disease state and macrophage activation [133].

Macrophage involvement in cardiac injury has also been seen in the setting of drug-induced cardiomyopathy, particularly in the case of doxorubicin. Doxorubicin is a well-known chemotherapeutic with documented cardiotoxicity, while doxorubicin-induced cardiomyopathy (DiCM) is a key contributor to heart failure and mortality in these settings [134]. Zhang et al. reported that in DiCM, macrophage infiltration into the cardiac tissue along with resident tissue macrophage proliferation contribute to the pathogenesis and resolution of the DiCM phenotype [119]. Indeed, in murine models, they report that doxorubicin initially attenuates the resident macrophage population and leads to a reactive proliferation, which then aids in repairing the DiCM [119]. Zhang et al. further noted that the ablation of SR-A1, a scavenger receptor thought to play a role in M2 macrophage proliferation, resulted in DiCM progression through the alteration of the expression levels of C-myc protein and an impaired restorative macrophage phenotype [119]. These therapeutics focused on these differences at the cellular level can potentially reduce mortality from DiCM and offer relief to patients [119].

## 7. Targeting Macrophages in the Treatment of CVD

Many of the potential macrophage-centered treatment options for CVD target pro-inflammatory cytokines that relate to macrophage-induced disease processes. The inhibition of IL-1β is a target being explored. IL-1β has been shown to promote the development of atherothrombotic plaque and macrophage adhesion [135]. In a study of 10,000 patients with histories of MI, the monoclonal anti-body drug canakinumab, which blocks the pathway of IL-1β, showed a significant reduction in cardiovascular events over a median follow-up period of 3.7 years [135]. The inhibition of IL-1β by canakinumab led to a reduction in heart-failure-related mortality in patients who had prior MI and active inflammation [136,137]. Another study found that the canakinumab treatment resulted in a dose-dependent 25% reduction in hospitalizations related to HF when treated with 300 mg canakinumab every three months [136]. IL-1β inhibition can offer a new approach to the treatment of HF that does not alter the blood pressure or renal function in a clinically significant way. Other therapeutic targets related to macrophage IL-1β expression include Rac-2-mediated pathways that play a role in atherosclerosis; however, no treatments have been developed for this target to date [138].

Interleukin-1 receptor blockade with anakinra preserves the LV systolic function after acute MI and attenuates systemic inflammation [46]. Anakinra may have a use in both HFrEF and HFpEF [46]. Anakinra demonstrated an improvement in peak aerobic exercise capacity and re-hospitalization of patients with heart failure (HF) when received for 12 weeks’ duration in comparison to treatment for 2 weeks’ duration or placebo [139]. This difference was not statistically significant but has created interest in exploring the IL-1 receptor blockade as a potential treatment option [139]. The left ventricular ejection fraction was also shown to increase with IL-1 blockade when compared to placebo-treated patients [140].

Interleukin-6 is produced by many cells in the cardiovascular system, including macrophages. IL-6 could also prove to be a promising therapeutic target. The IL-6 inhibitor tocilizumab was able to reduce the levels of pro-B-type natriuretic peptide in patients with rheumatoid arthritis and no CVD, implying that the drug may have cardioprotective properties [141]. Additionally, non-STEMI patients showed attenuated troponin T release and reduced systemic inflammation in the presence of tocilizumab, yet again demonstrating that IL-6 blockade’s effects may need to be further explored [142]. The blockade of IL-6 needs to further be explored in CVD. A further exploration of macrophage-expressed CCL2/CCR2 should also be considered [46,143]. The inhibition of CCL2/CCR2 raises concerns due to the macrophages’ role in both pro- and anti-inflammatory processes [137]. It is important to find ways to selectively discourage pro-inflammatory M1 macrophages, while encouraging M2 anti-inflammatory macrophage actions in the treatment of CVD processes [144,145].

Aside from interleukin-based strategies to target macrophages, other molecular pathways have had some success as well in various etiologies of CVD. The inhibition of protein phosphatase and tensin homolog (PTEN) with a vanadium-based compound called “VO-Ohpic” has been shown to attenuate inflammatory M1 macrophages and increase anti-inflammatory M2 macrophages, improving cardiac function in doxorubicin-induced cardiomyopathy [145]. Targeting cytokine MCP-1 may be a way to attenuate the M1 macrophage response in patients with dilated and uremic cardiomyopathy [146]. Baicalin, a plant-based anti-inflammatory flavonoid, has shown benefit in changing the macrophage phenotype from M1 to M2 in the treatment of CVD [120]. Additionally, another anti-inflammatory agent salvianolic acid B (SalB) has been shown to increase M2 macrophages and decrease M1 macrophages at 3 days, which was followed by reduced cardiac dysfunction at 7 days in a post-myocardial infarction mouse model [147].

The reduction in all macrophages was protective against myxomatous valvular disease progression in mouse models [148]. Refametinib and doxycycline were shown to reduce macrophage infiltration in mouse models with latent aortic valve disease, slowing its progression [146]. Evogliptin, an inhibitor of the enzyme dipeptidyl peptidase-4, was shown to attenuate aortic valve calcification through a reduction in pro-inflammatory cytokines (Il-6, IL-1, TNF-alpha) and the inhibition of macrophage infiltration in a rabbit model [149]. Evogliptin has also been shown to be useful as an anti-atherosclerosis therapy via its ability to inhibit vascular inflammation [150]. Studies using mouse models demonstrated that macrophage depletion by IV clodronate liposomes inhibited the development of hypertension, consequently reducing left ventricular hypertrophy, cardiac fibrosis, and cardiac remodeling [151]. Future trials are needed to explore the effects of denosumab and bisphosphonates on pro-inflammatory cytokines that lead to the development of calcific aortic valve disease [144].

The use of cardiosphere-derived cell (CDC) exosomes has proven to have some benefit for CVD in porcine and rat models. One study demonstrated a decrease in acute ischemic reperfusion injury and a decrease in chronic post-MI remodeling [152]. A two-fold reduction in macrophages was noted and groups that were treated with CDCs had lower left ventricular diastolic pressure, decreased lung congestion, and enhanced survival [152]. Clinical trials are underway testing the use of CDCs for the prevention of CVD [152]. The blockade of TNF-alpha with ethanercept and infliximab in HFrEF patients has been explored, and did not show any promise due to the complicated relationship between TNF-alpha and macrophages [46,137]. Macrophages and their associated pro-inflammatory cytokines are promising therapeutic targets that warrant further study in the treatment of CVD (Figure 4).

## 8. Conclusions

Macrophages play an extensive and complex role in maintaining the tissue homeostasis in the heart. Macrophages are now understood to display a diverse plasticity, depending on their resident tissue and developmental origin. Heart-specific macrophages are defined by the presence of the CCR2 receptors, and these specific cell types are intimately involved in the mechanisms that lead to the development of heart failure, atherosclerosis, and cardiomyopathy. Due to their involvement in CVD development, macrophages have shown promise as therapeutic targets to treat and prevent CVD conditions. Further research into the macrophage-related mechanisms of CVD and their clinical importance may provide potential novel strategies to combat the effects of CVD and reduce the substantial burden of CVD-related morbidity and mortality.

## Figures and Tables

**Figure 1 biomedicines-10-01579-f001:**
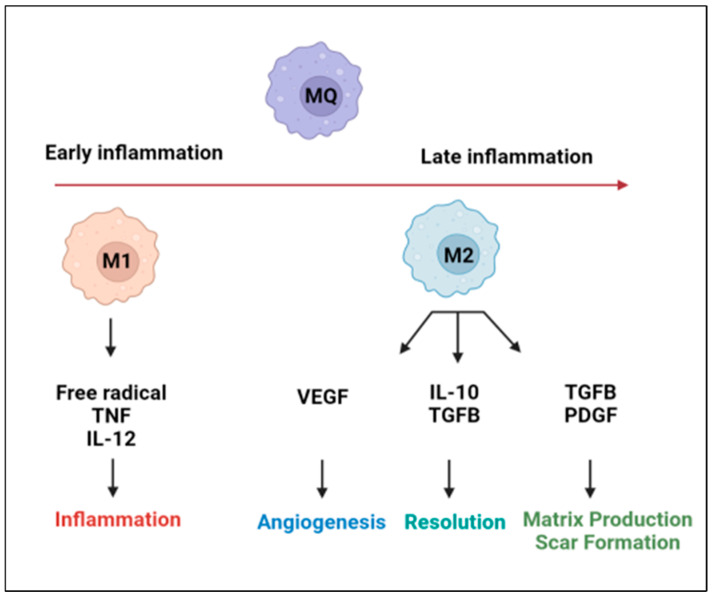
Schematic illustrating the inflammatory signaling mechanisms and the regulation of macrophages during early and late phases of inflammation. M1 macrophages are highly phagocytic and produce large amounts of pro-inflammatory mediators, whereas M2 macrophages produce large amounts of anti-inflammatory mediators and support angiogenesis and tissue repair.

**Figure 2 biomedicines-10-01579-f002:**
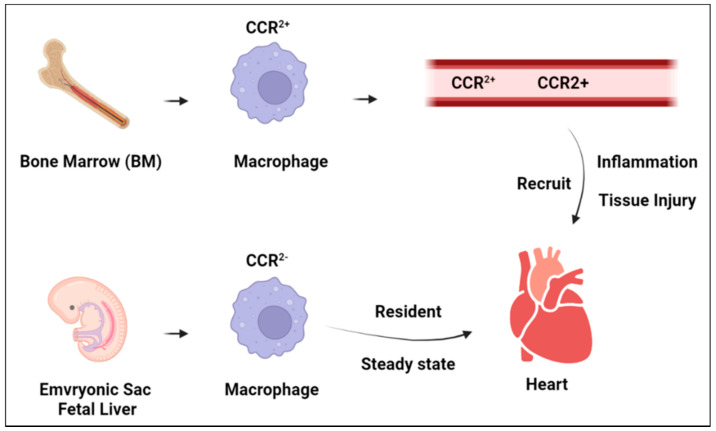
Schematic illustrating the function of CCR2^+^ and CCR2^−^ cardiac-resident macrophages in the heart.

**Figure 3 biomedicines-10-01579-f003:**
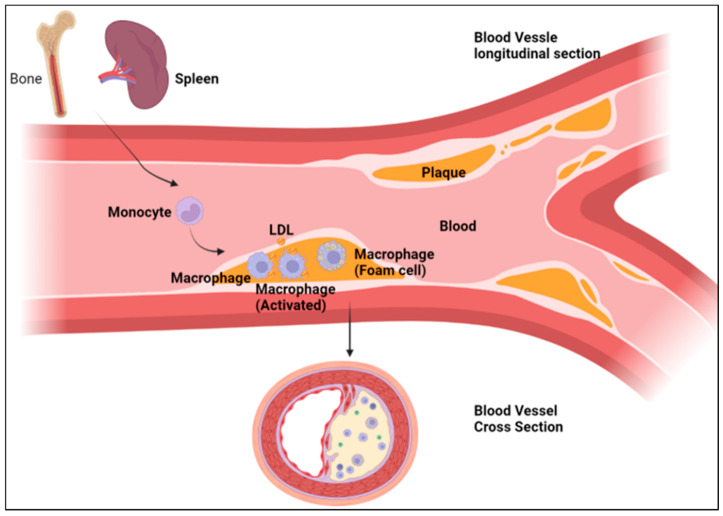
Schematic illustrating the role of macrophages in the development of atherosclerosis.

**Figure 4 biomedicines-10-01579-f004:**
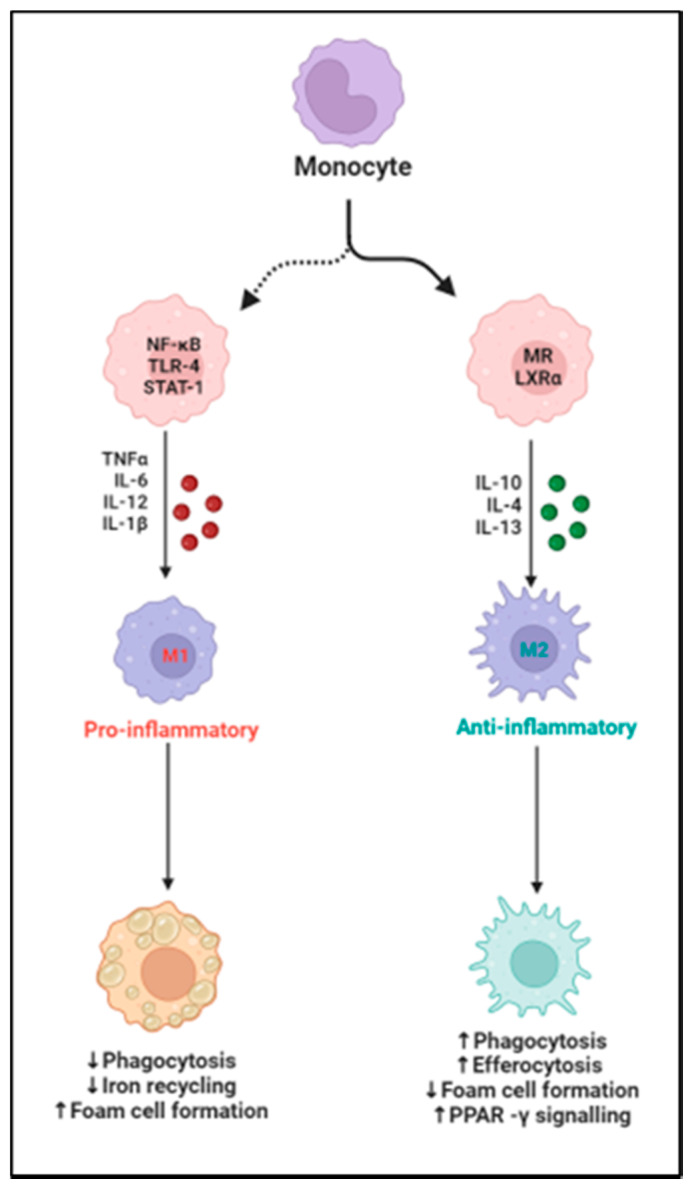
Schematic illustrating macrophage polarization to proinflammatory M1 and to an anti-inflammatory M2 phenotype and the specific factors affecting the process, up arrows mean increased levels, and down arrows refer to decreased levels.

## Data Availability

The datasets generated or analyzed during the current study are available from the corresponding author on reasonable request.

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
