# Peer review of "Dirty Jobs: Macrophages at the Heart of Cardiovascular Disease"

_biomedicines, 2022, doi:10.3390/biomedicines10071579_

Round 1

Reviewer 1 Report

The role of macrophages in cardiovascular diseases is explained based on their role in different heart disease conditions. It is a condensed research work of several research papers in the field of cardiovascular disease. The discussion about how the knowledge might be applied in developing new hypotheses and designing better studies to understand cardiovascular diseases is appreciated. Overall, the authors have done a decent job with the review.

Author Response

Reviewer 1

The role of macrophages in cardiovascular diseases is explained based on their role in different heart disease conditions. It is a condensed research work of several research papers in the field of cardiovascular disease. The discussion about how the knowledge might be applied in developing new hypotheses and designing better studies to understand cardiovascular diseases is appreciated. Overall, the authors have done a decent job with the review.

We thank the reviewer for their time to review the manuscript and positive comments.

Reviewer 2 Report

Reviewer comments and suggestions

The authors discussed the role of macrophages in association with the development of CVD. Macrophages play diverse roles in the maintenance of cardiovascular homeostasis and imbalance of associated mechanisms that may contribute to the development of CVD. In this review, the authors provided a diversity of macrophages, their roles in maintaining tissue homeostasis within the heart and vasculature, and how they regulate leads to CVD. The authors concluded the potential importance of macrophages in the identification of preventative, diagnostic, and therapeutic strategies for patients with CVD.

Based on my view, below are the comments that need to be incorporated into the revised version of the manuscript. 

  1. In the abstract, Line 19 which mechanism the authors want to mention here 
  2. Page 2 Line 17 such as, better to explore here
  3. In figure 1 the legend should be added here as well.
  4. Page 2 Line 24-26 already discussed above., please delete the lines
  5. Page 3 line 4 Better to explore the sentence with the help of cited references
  6.  Page 4 one reference is not enough “valvular heart disease, high blood pressure, and atherosclerosis [36]”. Same issue with the reference number 41
  7. How the authors could you elaborate on this “Macrophages are known to be homeostatic mediators in the heart and have demonstrated a substantial role in these mechanisms”
  8. Page 5, First time used here HRpEF  
  9. Some mistake in the structure of the sentence, please modify it “excess repetition of this process can lead to over[1]compensatory mechanisms that thin the myocardial wall and chambers and can lead to chronic HRrEF”
  10. Page 6 the sentence was repeated in the above paragraph “such as the release of IL-10, has been linked to the excess collagen deposition and myocardial stiffening over time”.
  11. Page number 6 Line 17-19 is this the only reason for this 
  12. What do the authors want to say here, please explore, “specific targeting of the expression of chemokine receptors and cell adhesion molecules on monocytes in their mechanism of macrophage differentiation is a rapidly developing area of research”
  13. If there were already reviews on the same topic so what is the matter of writing this section? it would be better not to include as such statement “on the mechanism of macrophages in atherosclerosis have been well described in the following reviews[80, 81]”
  14. Reference 119 What do the authors want to state here and page number 9 (POH) first time used
  15. Page number 10 How it comes cancer discussion here.
  16. Please check typo error in the section “ Targeting Macrophages in the Treatment of CVD”
  17. Is the data was from CVD or any other disease “blockade of IL-6 needs to further be explored in CVD. Further exploration of macrophage-expressed CCL2/CCR2 should also be considered [43, 141]. Inhibition of CCL2/CCR2 raises concerns”
  18. The spelling of anti-inflammatory is wrong inside figure 4
  19. “CCR2 receptors”, one time used this word, why the authors did not discuss in the text-only in conclusion part
  20. Please check the format of references used by MDPI Journals, the authors have to change all.

Reviewer 3 Report

The authors reviewed the role of macrophages in cardiovascular disease. The review is well written. However, reading the document in its introductory part relating to the typization of the cellular type is missing a section dedicated to the techniques of visualization. I refer in particular to the classical morphology of the macrophage including light and electron microscopy, the identification of the various antigens and subsequently  the important problem of polarization with the identification of the various populations and their function. I believe that the deepening of this part can also be of valid help for the reader outside the research field but who can draw valid help for other types of problems.

Author Response

Reviewer 3

The authors reviewed the role of macrophages in cardiovascular disease. The review is well written. However, reading the document in its introductory part relating to the typization of the cellular type is missing a section dedicated to the techniques of visualization. I refer in particular to the classical morphology of the macrophage including light and electron microscopy, the identification of the various antigens and subsequently the important problem of polarization with the identification of the various populations and their function. I believe that the deepening of this part can also be of valid help for the reader outside the research field but who can draw valid help for other types of problems.

We thank the reviewer for their time to review the manuscript. We have added references to two review papers in the introduction section that discuss the use of electron microscopy in identifying the structure and function of macrophages. This will point out the role of microscopy in the typization of macrophages to the readers.

Round 2

Reviewer 3 Report

The authors answered at my questions